# PML/RARa Interferes with NRF2 Transcriptional Activity Increasing the Sensitivity to Ascorbate of Acute Promyelocytic Leukemia Cells

**DOI:** 10.3390/cancers12010095

**Published:** 2019-12-30

**Authors:** Cristina Banella, Gianfranco Catalano, Serena Travaglini, Mariadomenica Divona, Silvia Masciarelli, Gisella Guerrera, Francesco Fazi, Francesco Lo-Coco, Maria Teresa Voso, Nelida Ines Noguera

**Affiliations:** 1Department of Biomedicine and Prevention, Tor Vergata University of Rome, 00133 Rome, Italy; cristina.banella@gmail.com (C.B.); gianfranco.catalano@uniroma2.it (G.C.); serenatravaglini@live.it (S.T.); francesco.lo.coco@uniroma2.it (F.L.-C.); Voso@med.uniroma2.it (M.T.V.); 2Neuro-Oncohematology Unit, Fondazione Santa Lucia, Istituto di Ricovero e Cura a Carattere Scientifico (I.R.C.C.S.), 00143 Rome, Italy; 3Oncologia Medica, Policlinico Universitario Tor Vergata, 00133 Rome, Italy; mariadomenica.divona@ptvonline.it; 4Istituto di Istologia ed Embriologia, Universita Cattolica del Sacro Cuore, 00168 Rome, Italy; silvia.masciarelli@uniroma1.it; 5Fondazione Policlinico Universitario A. Gemelli, I.R.C.C.S., 00168 Rome, Italy; 6Department of Anatomical, Histological, Forensic & Orthopedic Sciences, Sapienza University of Rome, Laboratory Affiliated to Istituto Pasteur Italia-Fondazione Cenci Bolognetti, 00161 Rome, Italy; francesco.fazi@uniroma1.it; 7Neuroimmunology and Flow Cytometry Units, Fondazione Santa Lucia I.R.C.C.S., 00143 Rome, Italy; g.guerrera@hsantalucia.it

**Keywords:** acute promyelocytic leukemia, PML/RARa, NRF2, *HMOX1*, ascorbate, ROS

## Abstract

NRF2 (NF-E2 p45-related factor 2) orchestrates cellular adaptive responses to stress. Its quantity and subcellular location is controlled through a complex network and its activity increases during redox perturbation, inflammation, growth factor stimulation, and energy fluxes. Even before all-trans retinoic acid (ATRA) treatment era it was a common experience that acute promyelocytic leukemia (APL) cells are highly sensitive to first line chemotherapy. Since we demonstrated how high doses of ascorbate (ASC) preferentially kill leukemic blast cells from APL patients, we aimed to define the underlying mechanism and found that promyelocytic leukemia/retinoic acid receptor α (PML/RARa) inhibits NRF2 function, impedes its transfer to the nucleus and enhances its degradation in the cytoplasm. Such loss of NRF2 function alters cell metabolism, demarcating APL tissue from both normal promyelocytes and other acute myeloide leukemia (AML) blast cells. Resistance to ATRA/arsenic trioxide (ATO) treatment is rare but grave and the metabolically-oriented treatment with high doses of ASC, which is highly effective on APL cells and harmless on normal hematopoietic stem cells (HSCs), could be of use in preventing clonal evolution and in rescuing APL-resistant patients.

## 1. Introduction

Acute myeloid leukemias (AMLs) are clonal disorders of the hematopoietic system characterized by the rapid growth of abnormal precursors cells that interfere with normal hemopoiesis. Acute promyelocytic leukemia (APL) is a subtype of AML characterized by a reciprocal and balanced translocation involving retinoic acid receptor α (RARa) on chromosome 17 and promyelocytic leukemia (PML) on chromosome 15 which generates the oncogenic PML-RARα fusion protein. Central in the pathogenesis of APL, PML/RARa impairs formation of functional PML nuclear bodies, and acts as a transcriptional repressor antagonizing myeloid differentiation, altering DNA repair and promoting the self-renewal capacity of APL-initiating cells [1,2,3,4].

Cap’n’Collar basic leucine zipper transcription factor NF-E2 p45-related factor 2 (NRF2) plays a crucial role in orchestrating adaptive cell response to stress. NRF2 binds to antioxidant response elements (AREs) in the promoter region of genes involved in redox regulation, proteostasis, DNA repair, apoptosis, nutrient and xenobiotic metabolism [5,6,7,8,9,10,11].

Ascorbic acid, commonly known as an antioxidant when used in low doses, is an essential nutrient that acts as a coenzyme in several enzymatic reactions. In the early 1970s, Pauling and Cameron reported on the possible benefit of high-dose intravenous and oral ASC for cancer patients [12,13]. A recent study showed that high-dose vitamin C selectively kills colon cancer cells carrying KRAS (KRAS proto-oncogene, GTPase) or BRAF (B-Raf proto-oncogene, serine/threonine kinase) mutations [14]. A series of in vitro and in vivo studies revived the interest of high doses of ascorbate (ASC) as an anticancer agent. In particular, studies from our group and others revealed that, when administered in pharmacologic doses corresponding to the millimolar (mM) range, ASC behaves as a powerful pro-oxidant, generating hydrogen peroxide-dependent cytotoxicity towards a variety of cancer cells in vitro without adversely affecting normal cells [15,16,17,18]. A different view of the potential action of vitamin C in cancer came from the discovery of its importance for the activation of the ten-eleven translocation (TET) and Jumonji dioxygenases that are involved in active demethylation of DNA and histones, respectively [19,20].

NRF2 is tightly regulated to assure cell homeostasis: In normal conditions low levels of NRF2 are maintained by the association with Kelch-like ECH-associated protein 1 (Keap1) in a complex with E3 ubiquitin ligase Cullin 3-Ring-box protein 1 (CUL3-RBX1) and degradation by proteasome [21,22]. Oxidants and electrophilic molecules induce modifications in Keap1 that prevents the ubiquitination of NRF2 [23]. NRF2 then translocates to the nucleus, heterodimerizes with small musculoaponeurotic fibrosarcoma (sMAF) proteins and binds to antioxidant response elements (AREs) to promote the expression of more than 200 genes [24,25,26]. Finally, it entangles with a Skp1-Cullin 1 (CUL1) F-box-containing complex that targets NRF2 for proteasomal degradation or with RNF4 E3 ubiquitin ligase leading to degradation in promyelocytic-nuclear bodies (PML-NB) domains within the nucleus [27,28,29,30]. Hematopoietic cells appear to be particularly vulnerable to the accumulation of reactive oxygen species (ROS), since deficiencies in ROS scavenger proteins result in severe anemia and/or carcinogenesis [31,32]. Thus the tight regulation of NRF2 is important to assure cell homeostasis. Stem cells have a low energetic profile with scarce ROS production. In the hemopoietic compartment, the loss of NRF2 results in defective differentiation and decreased survival as well as defective engraftment after hematopoietic stem cell (HSC) transplantation [33]. However, there is no increased ROS production but rather a sensitization to ROS hyper-production induced by oxidative stress [33]. Since we demonstrated that high doses of ascorbic acid (ASC) preferentially kill leukemic blasts from APL patients while sparing normal HSCs [17,18] we aimed to define the effect of PML/RARa on NRF2 function in hematopoietic cells.

## 2. Material and Methods

### 2.1. Primary Patient Samples

Bone marrow (BM) mononuclear cells (MNC) were collected from patients with newly diagnosed AML and APL. APL diagnosis was confirmed in all cases through detection of the PML/RARa fusion gene by RT-PCR. BM-infiltration by leukemic blasts was >70 % in all patients included in the study. BM-MNC isolated from healthy individuals were used as control. Written informed consent was obtained from all patients according to institutional guidelines and the declaration of Helsinki. The study had been approved by the independent ethical commitment of Foundation PTV Policlinico Tor Vergata—Experimental Register R.S. 208/18 on 23th January 2019.

Total RNA was extracted from Ficoll-Hypaque-isolated mononuclear cells using the method of Chomczynsky and Sacchi. RNA was reverse-transcribed using random hexamer primers.

### 2.2. Cell Cultures

NB4, an APL-derived cell line, carrying the t(15;17) translocation was purchased from DSMZ (Braunschweig, Germany); U937, an histiocytic lymphoma-derived cell line; Mock and PR9 (a zinc-inducible PML/RARa model constructed from the U937 cell line) [34]; and HL-60, an AML FAB M2-derived cell line and Oci-AML2, an AML-M4-derived cell line carrying the DNMT3A R635W mutation (kindly provided by Emanuela Colombo, European Institute of Oncology, Milan, Italy) were grown at 37 °C in a humidified atmosphere of 5% CO2 in air in RPMI (Roswell Park Memorial Institute Medium) (GIBCO-BRL, Grand Island, NY, USA) supplemented with 10% fetal bovine serum (FBS) (GIBCO-BRL), 20 mM Hepes, 100 μg/mL penicillin and 100 μg/mL streptomycin (GIBCO-BRL). 

The HEK293T, a human embryonal kidney cell line (kindly provided by Corinna Giorgi, European Brian Research Institute, Rome) was cultured as a monolayer in Dulbecco’s modified eagle medium containing 10% fetal bovine serum, 100 mg/mL penicillin and 100 mg/mL streptomycin.

Cells were tested regularly for mycoplasma contamination using a PCR kit (N-Garde EMK090020, Euroclone, Milan, Italy).

### 2.3. Immunofluorescence Assays

Immunofluorescence was used to evaluate the localization of NRF2 in the presence of PML/RARa. U937-PR9 and Mock control cells were prepared using a cytocentrifuge. Assays were performed as previously described [35]. Briefly, cells fixed with 4% paraformaldehyde (PFA) were permeabilized in PBS containing 0.1% Nonidet P-40 and blocked in 3% bovine serum albumin (BSA) (Sigma-Aldrich, St. Louis, MO, USA). Slides were incubated overnight with the primary antibody anti-PML, kindly provided by Brunangelo Falini; and anti-NRF2 (Ab Cam, Cambridge, UK); and following two PBS washes and incubated for 2 h with the secondary antibodies: Alexa Fluor 555-labeled goat anti-mouse and Alexa Fluor 488-labeled goat anti-rabbit (diluted 1:400 with PBS+BSA 3%) (Invitrogen, Carlsbad, CA, USA).

The nuclei were stained with 1 μg/mL DAPI (4′,6-diamidino-2-phenylindole) (Sigma-Aldrich, St. Louis, MO, USA) for 5 min in PBS. Finally, cells were rinsed and mounted in Fluoromount (Sigma-Aldrich). Images were acquired using a Zeiss LSM 700 (Carl Zeiss Microscopy, Jena, Germany) confocal laser scanning microscope.

### 2.4. Quantitation of ROS

To evaluate ROS levels cells were incubated for 24 h with Zn. The Abcam’s ROS assay kit “ab113851” (Abcam, Cambridge, UK) uses the cell permeant reagent 2′,7′-dichlorofluorescein diacetate (DCFDA), a fluorogenic dye that measures hydroxyl, peroxyl and other ROS activity within the cell. After diffusion into the cell DCFDA was deacetylated by cellular esterases to a non-fluorescent compound, which was later oxidized by ROS into 2′, 7′ –dichlorofluorescein (DCF). The cells were then analyzed using a Beckman Coulter CyAn ADP (Beckman Coulter, Chapel Hill, NC, USA).

### 2.5. Transfection Experiments

To evaluate the effect of PML/RARa on NRF2 half life, Hek293T cells were transfected with pSG5-PML/RARa or pSG5 constructs using jetPRIME^®^transfection reagent (Polyplus-transfection^®^, Strasbourg, France). The cells were treated with ZnSO_4_ 100 µM for two hours to induce NRF2 expression and then the stops were initiated over five hours. 

HL-60 cells were transfected with pSG5-PML/RARa or pSG5 constructs using Amaxa Cell Line Nucleofector Kit C (Lonza, Cologne GmbH, Germany). After 24 h the cells were treated with ZnSO4 100 µM for two hours to induce NRF2 expression and then with cycloheximide (CHX) (Sigma-Aldrich, St. Louis, MO, USA), 0.2 nM for another four hours.

### 2.6. Western Blot Analysis

Western blot was used to evaluate the expression level of NRF2 and Keap1 [36]. Cell pellets were re-suspended in lysis buffer containing 10 mM Tris-HCl (pH 7.4), 5 mM etilendiammonotetracetico (EDTA) (Sigma-Aldrich, St. Louis, MO, USA)., 150 mM NaCl, 1% Triton X-100, 250 μM orthovanadate, 20 mM β-glycerophosphate and protease inhibitors (Sigma-Aldrich). Lysates were centrifuged at 10000 g for 15 min at 4 °C and supernatants were stored at −80 °C. Twenty microgram aliquots of proteins were resuspended in a reducing Laemmli Buffer (with β-mercaptoethanol) and loaded onto a 10% polyacrylamide gel, then transferred to nitrocellulose membrane. After blocking with 5% milk (Fluka, Sigma-Aldrich, St. Louis, MO, USA), the membranes were incubated with specific antibodies, as listed in Appendix A. IgG-Horseradish peroxidase-conjugated preparations were used as secondary antibodies and the enhanced chemiluminescence (ECL) procedure was employed for development (ECL kits, Amersham, Buckinghamshire, UK). Nuclear and cytoplasm fraction were obtain using NE-PER nuclear and cytoplasmic extraction reagents from Thermo Scientific (Rockford, IL, USA) according to instructions [36].

The autoradiograms obtained were scanned and exported for densitometry analysis. Protein signal intensities were measured by using ImageJ-win64 software.

### 2.7. Quantitative Real-Time Expression Analysis

Q-RT-PCR was used to evaluate the expression level of *NRF2* and its target genes *HMOX1*, *NQO-1* and *AKR1C1*. Reverse transcription was performed using 1 μg total RNA and a standardized protocol (Applied Biosystems, Foster City, CA, USA). The primers were obtained from Sigma (Saint Louis, MO, USA) (Appendix A). The reaction mixture of 20 µL contained 1× Syber Green Supermix (Biorad, Benicia, CA, USA), 300 nM of each primer. All expression levels (gene of interest and normalization control) in quantitative RT–polymerase chain reactions (RQ-PCR) were obtained by the use of standard curves. Primers used are shown in Appendix A.

### 2.8. Survival Assay

To study if PML/RARa affected the cell sensitivity to ASC survival assays were performed. Cells were plated in 96-well plates at 7000 cells/well, exposed to vehicle, 1 mM or 3 mM ASC, and cultured for three days. Cell survival was evaluated by the CellTiter 96 ^®^ AQueous One Solution Cell Proliferation Assay Kit (Promega, Boston, MA, USA) or ATPlite Luminescence Assay System (Perkin Elmer GmbH, Überlingen, Germany) according to the manufacturer’s instructions.

### 2.9. Immunoprecipitation

To analyze if physical interaction between PML/RARa and NRF2 existed, total extracts and nuclear and cytoplasm fractions were used for co-immunoprecipitation assays. Briefly, 1 mg of extract was incubated overnight at 4 °C with 2 μg of antibody (anti-NRF2, anti-PML and anti-RARa), and subsequently for 45 min at 4 °C with Dynabeads Protein G (Invitrogen, Dynal AS, Oslo, Norway). Samples were then blotted with anti-NRF2 or anti-RARa antibodies (Appendix A).

### 2.10. CHIP Assay

To analyze if PML/RARa affected the binding of NRF2 to its target genes, experiments on chromatin immunoprecipitation (ChIP) assays were performed. The genomic regulatory regions for HMOX1 gene are located upstream the first exons of the HMOX1 locus (10 kbs, position: 35371096 to 35381173). Reichard et al. demonstrated that NRF2 binds to two distal ARE enhancer sites far upstream of the HMOX1 transcription start site (TSS) [37,38]. For this reason, quantitative real-time PCR (qPCR) was performed targeting the distal antioxidant response element (ARE) of the HMOX1 gene. ChIP assays were performed as previously described [39]. Briefly, DNA was double-crosslinked to proteins with 1% formaldehyde (Sigma, St Louis, MI, USA) and after incubation for 20 min at room temperture, glycine was added to a final concentration of 0.125 M, for five minutes. After washing twice in 1 × PBS, cell lysis buffer (10 mM Tris pH 8.0, 100 mM NaCl and 0.2 % NP40) was added to the samples, and cells were incubated on ice for 30 min. Nuclei were pelleted at 1500 rpm at 4 °C, and after addition of the nuclear lysis buffer (50 mM Tris pH 8.1, 10 mM EDTA and 1 % sodium dodecyl sulfate (SDS), were incubated on ice for 30 min. Chromatin fragments of around 200–300 bp were obtained by sonication, using a Branson Sonifier 450 Analog Cell Disruptor. For each immunoprecipitation, 3 mg of antibodies (Appendix A) were conjugated to magnetic beads (G-protein magnetic beads, Invitrogen, Dynal, Oslo, Norway). After extensive washing, bound DNA fragments were eluted and analyzed by quantitative PCR using the SYBR Green Master Mix (Biorad, Benicia, CA, USA). ChIP signals were normalized against the input and expressed as relative enrichment of the material, precipitated by the NRF2 antibody binding to the HMOX1. Relative quantification using the comparative Ct method (percent input method, signals obtained from the ChiP divided by signal from an input sample adjusted to 100%). In addition, an unrelated sequence in the GAPDH gene was used as a negative control (primers listed in Appendix A).

### 2.11. Statistical Analysis

All statistical analyses were conducted using the GraphPad Prism5 software [40]. The unpaired *t*-test compares the means of two unmatched groups, assuming that the values follow a Gaussian distribution; when the distribution was not normal we used the Mann–Whitney test, and all tests were done on two sides.

## 3. Results

### 3.1. NRF2 Protein Level, but Not mRNA, Is Lower in APL Than in Other AML

Since in a previous work we demonstrated a peculiar sensitivity of APL blast cells to pharmacologic doses of ascorbate [17] we decided to examine NRF2 expression in these cells because of the important role of NRF2 in orchestrating cellular response to stress. Using western blot, we found that APL blasts express NRF2 protein at a significantly lower level (0.6 ± 0.8, *n* = 8) than other AMLs samples (1.2 ± 0.4, *n* = 7) (*p* = 0.02) (Figure 1a). On the other hand, *NRF2* mRNA expression, assessed by Q-RT-PCR, was significantly higher in APL (*n* = 13; 0.12 ± 0.1) and in AML (*n* = 12; 0.24 ± 0.3) as compared to normal bone marrow (NBM) (*n* = 5; 0.03 ± 0.01, APL vs. normal bone marrow (NBM), *p* = 0.04, AML vs. NBM, *p* = 0.004) (Figure 1b), indicating post-transcriptional regulation of NRF2 expression. To note that Keap1 protein, the main regulator of NRF2 degradation, is evenly expressed in APL *n* = 8 (0.91 ± 0.3) and AML *n* = 7 (0.99 ± 0.2) patients’ samples (Figure 1c), hence a different player is involved.

### 3.2. NRF2 Transcriptional Activity Is Inhibited in APL Cells

To ascertain NRF2 transcriptional activity in APL cells we measured mRNA expression of three NRF2 target genes in cells isolated from the bone marrow of 13 APL patients, 12 AML patients and five healthy donors (NBM) using quantitative RT-PCR. We tested *HMOX1*, *NQO-1* and *AKR1C1*: In APLs mean values were: 1.7 ± 1.8; 10.5 ± 9.8 and 0.25 ± 0.37; in AMLs: 30.6 ± 30.2; 44 ± 46 and 0.9 ± 1.8, and in NBMs: 1.6 ± 1.8; 1.1 ± 0.5 and 1.0 ± 0.3 respectively. AML samples were characterized by heterogeneous expressions levels, average expression was significantly higher than that of APL samples (*p* = 0.0007; *p* = 0.0016 and *p* = 0.005 respectively) (Figure 2a) clearly showing inhibition of NRF2 transcriptional activity in APL cells.

### 3.3. PML/RARa Inhibits the Increase of NRF2 Protein and Interferes with NRF2 Transcriptional Activity by Preventing Its Binding to ARE Motifs

To study the effect of PML/RARa expression on NFR2 we used a myeloid-inducible system: PR9 cells (U937 cell line with a zinc inducible PML/RARa expression) and Mock control cells (U937 lacking PML/RARa sequence on the transfected construct). Since ZnSO_4_ induces NRF2 expression we felt it was a good system to analyze PML/RARa interference with NRF2 induction. After 100 µM zinc sulphate (ZnSO_4_) addition, in Mock cells *NRF2* mRNA increases after six hours (1.5 ± 0.4) and slightly decreases in the following hours, whereas in PR9 cells the increase is higher in the first six hours (2.46 ± 1) and persistent up to 24 h (Figure 2b). Conversely NRF2 protein levels augment in Mock cells more than four folds in the first six hours (2.1 ± 0.1) and persists two folds higher after 24 h (1 ± 0.8), whereas in PR9 cells, after a slight increase in the first three hours (1 ± 0.4), the protein abates to a level lower than normal (0.5 ± 0.1) (Mock vs. PR9 *p* = 0.008) up to 24 h (Figure 2c), suggesting that PML/RARa doesn’t interfere with NRF2 mRNA induction but with the protein stability. To further study if that down-modulation of NRF2 affects the expression of its target genes we measured in the same system the expressed levels of *HMOX1*, that normally exerts an important antioxidant effect in cancer cells. We could see that in PR9 cells after an initial induction at three hours, much lower than the control’s (11 ± 0.5 vs. 32 ± 8), and transcriptional activity drops (PR9 cells + Zn 1.8 ± 0.6 vs. Mock + Zn 42.7 ± 11 at 6 h, *p* = 0.003) (Figure 2d). One should note that the kinetics of the NRF2 protein and the induced *HMOX1* expression are identical and indicates that the lack of NRF2 function in the presence of PML/RARa stimulates its transcription from the homeostatic cellular apparatus to no avail, since there is no increase in NRF2 protein and function.

To ascertain the specificity of PML/RARa inhibition we performed a specific ChIP analysis on distal *HMOX1* ARE motifs [29,30] (Figure 2e) using the inducible system PR9 and Mock control cells. After four hours of treatment with ZnSO_4_ the binding of NRF2 to *HMOX1* gene is seven folds higher in Mock cells compared to PR9 cells (12 ± 8 vs. 1.7 ± 1.1; *p* = 0.02) (Figure 2f) demonstrating the specific inhibition of NRF2 physical binding to the target ARE sequence in the presence of PML/RARa.

### 3.4. PML/RARa Binds to NRF2, Impairs Its Nuclear Translocation, Promotes Its Cytoplasmic Degradation and Shortens Its Lifespan

Since the transcriptional function of NRF2 is nuclear we decided to look into the PML/RARa effect on the cellular localization of NRF2. By confocal microscopy we observed that, in untreated and un-induced PR9 cells, the NRF2 protein was localized mainly in the nucleus in PML-NBs. Eight hours after PML/RARa induction, in coincidence with PML NBs disruption, NRF2 protein was mainly confined to the cytoplasm (Figure 3a). We confirmed the subcellular localization of NRF2 protein by western blot analysis, separating nuclear and cytoplasmic cellular fractions. In Mock control cells we registered a 50% enhancement of nuclear NRF2 (*p* = 0.01), conversely in PR9 cells PML/RARa expression caused a 26% enhancement of cytoplasmic NRF2 (*p* = 0.009) (Figure 3b). 

We performed co-immunoprecipitation experiments, demonstrating a physical interaction between the two proteins as part of the mechanism for the inhibition of NRF2 transcriptional activity (Figure 3c). Since in the PR9 system PML/RARa protein localize mostly in the nucleus according to microscopy (Figure 3a) we investigated if there was a physical interaction between PML/RARa and NRF2 in both the cytoplasm and the nucleus. We performed a co-immunoprecipitation assay on fractioned nuclear and cytoplasmic extracts using an anti-NRF2 antibody and challenged the blot with anti-RARa antibody, demonstrating a physical interaction in both fractions (Figure 3d). In the experimental conditions there is a larger amount of complexed PML/RARa/NRF2 proteins in the nuclear fraction in line with the visual impression from the immunofluorescence analysis (Figure 3a). That probably indicates that PML/RARa inhibits NRF2 action by binding it to form a complex in the nucleus.

To investigate PML/RARa effect on NRF2 protein quantity we treated transfected Hek293T cells with PsG5-PML/RARa or control PsG5 constructs with ZnSO_4_. In PML/RARa-transfected cells, notwithstanding the oxidative stress, NRF2 expression abated by half in one hour; conversely, in the controls the expression increased by 1.6 times at the same time point (*p* = 0.001), and continued to increase up to 2.3 times at five hours while in PML/RARa-transfected cells it remained constant (Figure 3e). 

NRF2 is a rapid turnover protein with a limited life span, so to see if PML/RARa affected the NRF2 life span we treated PR9 and Mock cells with ZnSO_4_, then with the protein biosynthesis inhibitor cycloheximide (CHX) (100μg/mL) (Sigma Aldrich, Germany). PML/RARa-bearing cells exhausted NRF2 allowance in less than two hours, whereas in control cells the stress due to the experimental condition preserved a considerable quantity of the protein (Figure 3f). Thus PML/RARa significantly reduced NRF2 protein, at least in part, by reducing its half life. A similar result was obtained treating the myeloid HL-60 cell line transfected with PsG5-PML/RARa or control PsG5 constructs with ZnSO_4_ for two hours and then with CHX (100μg/mL) for four hours. In the control cells NRF2 expression was much higher respect to PML/RARa transfected cells, the difference persisted after CHX treatment (Figure 3g).

### 3.5. PML/RARa Expression Sensitizes Cells to Ascorbate Treatment

To evaluate NRF2 inhibition by PML/RARa on the oxidative stress defense mechanism of the cell we measured the levels of ROS content in NB4 and U937 cell lines and PR9 and Mock cell lines systems after induction with ZnSO_4_. As mentioned before zinc sulphate induces oxidative stress, and enhances NRF2 protein content, but in PR9 cells it induces PML/RARa expression as well. To further explore the oxidative unbalance, we respectively added 0.5 and 1 µM of rotenone/antimycin A and 5 µM Carbonyl cyanide p-(tri-fluromethoxy) phenyl-hydrazone (FCCP) and growing concentration of ASC (1.3 and 5 mM), to induce oxidative stress. In Mock cells zinc sulphate addition induced NRF2 expression and ROS cellular content remained stable, but PML/RARa expression in PR9 cells with consequent NRF2 loss of function caused significant augment of ROS (Figure 4a). Challenged with the other oxidants both cell line systems showed that NRF2 promptly responded to the challenge when PML/RARa is not present; conversely, its presence gravely impairs NRF2 homeostatic function (Figure 4a,b).

To confirm that NRF2 inhibition is involved in the mechanism of enhanced sensitivity to the ascorbate of APL cells we treated the two different cellular systems with ASC 1 mM and measured NRF2 and HO-1 protein expression and found that the presence of PML/RARa downgraded NRF2 protein quantity and downregulated HO-1 protein (Figure 4c,d). To make sure that the effect we saw was due to transcriptional deficiency of NRF2 we measured *HMOX1* mRNA levels in the NB4/U937 system by RT-PCR after treatment with ASC 1mM for 24 h. We observed a clear inhibition of *HMOX1* transcript induction in NB4 cells as compared with control cells (2.3 ± 3.2 vs. 87.3 ± 8.2 at 3 h and 2.9 ± 2.9 vs. 59.4 ± 13.8 at 6 h) *p* = 0.005 (Figure 4e). To confirm our finding in a natural occurring system we then challenged with ASC 3 mM for 24 h two sets of fresh primary blast, one from an APL patient and one from a patient with a different AML subtype, which confirmed that NRF2 downgraded in the presence of PML/RARa (Figure 4f).

We had proof that both in cell lines and fresh blasts from patients the presence of PML/RARa inhibits the expression of NRF2 target genes, which should sensitize cells to treatment with ASC 1 mM. In accordance with our data showing inhibition of NRF2 activity by PML/RARa, we observed; by ATP Lite test in NB4 cells and MTT viability test in induced PR9 cells, that PML/RARa sensitized to ASC, registering an enhancement of its efficacy in the presence of PML/RARa protein (respectively *p* = 0.04 and 0.03) (Figure 4g,h).

## 4. Discussion

NRF2 unbalance leads to different but significant effects depending on the cell environment, its ability for suppressing carcinogenesis and preventing metabolic unbalance on one end but promoting cancer progression and resistance to chemotherapy on the other [41,42]. Stem cells have a low energetic profile with scarce ROS production and in the hematopoietic compartment loss of NRF2 results in defective differentiation and decreased survival as well as defective engraftment after hematopoietic stem cell (HSC) transplantation. It does not cause enhancement of ROS production as in most tissues but rather a sensitization to ROS hyper-production with increasing rates of spontaneous apoptosis and decreased survival when exposed to oxidative stress [33,43]. Consequently, NRF2 malfunction in APL-initiating stem cells is bound to expose them to redox unbalance in the presence of oxidizing agents.

Different cross-regulations link ROS production, NRF2 and the molecular mechanisms involved in APL: In physiological conditions both partners of the hybrid protein (PML and RARa) interfere with NRF2 regulation and function. In particular, nuclear and cytoplasmic PML acts as a stress sensor and minimizes ROS accumulation. In normal cells the loss of PML induces ROS production and NRF2 import to the nucleus, and increases NRF2 protein abundance and stability. PML directly facilitates ROS scavenging and, in unstressed cells, indirectly contains NRF2 accumulation and trans-activating capability both in the cytoplasm and in PML-NBs [28,44]. Retinoic acid receptor alpha (RARa) agonists, as all-trans retinoic acid (ATRA), impair induction of ARE-driven genes by NRF2 and RARa binds directly to NRF2 protein [45,46,47]. Studying the effect of treatment with megadoses of ascorbate in AML blast cells, we noticed as APL cells were particularly sensitive [17]. Here we describe that the impaired function of NRF2, the true orchestrator of redox balance and stress response, is at the base of that finding.

In the NB4 cell line ASC treatment, while reducing PML/RARa cellular content, causes APL phenotype reversion with the reassembly of PML-NBs, as observed by a confocal microscopy [17]. Following PML/RARa expression and PML NBs disruption in the PML/RARa-inducible cells PR9, NRF2 protein is clearly confined to the cytoplasm (Figure 3a). As demonstrated in several studies even if PML-RARα is mainly nuclear as PML, it localizes in both nucleus and cytoplasm [48]. Lin et al. describes that the cPML/RARα disrupts cPML-Smad2/3 interaction and antagonizes the tumor suppressive TGF-β signaling, providing an additional mechanism for PML/RARα oncogenic function [49]. Interestingly, cPML mutants interact with and stabilize the PML/RARα cytoplasmic complex, resulting in potentiating of the PML/RARα oncogenic function [50]. Another study by Giorgi et al. reported the enrichment of cPML at the endoplasmic reticulum (ER) and the mitochondria-associated membranes (MAMs) mediating apoptosis responses upon various stimuli [51]. By immunoprecipitation experiments we could see that PML/RARa interacts with NRF2 mainly in the nucleus, but we also see interaction outside the nucleus and we cannot exclude that PML/RARa could affect NRF2 function also by acting in the cytoplasm. The nuclear interaction may help in impairing NRF2 transcriptional activity by preventing its binding to the DNA responding elements. An inhibitory direct interaction between the Neh7 domain of NRF2 and receptors RARa and RXRa [45,47,52] has been described. PML/RARa could exert a similar action, since we demonstrate by CHiP analysis that in the presence of PML/RARa, NRF2 binding to the target ARE sequences is inhibited, but of course the same domain could act differently in the contest of the chimeric protein, and we did not investigate this. In APL primary cells, the basal NRF2 protein level is higher with respect to NBM, but its quantity and function is lower compared to other AML subtypes, affecting stem cells homeostasis and exposing tumor cells to redox unbalance.

Our data depict a new mechanism determined by the presence of PML/RARa in APL tumor cells involving NRF2. Contrary to a previous report on the short form of PML/RARa ectopically transfected in breast cancer MCF7 cells [47], our data show that in primary leukemic cells and in leukemic and non hematologic cell lines NRF2 action is hampered by PML/RARa: Its life span and transcriptional function are indeed highly reduced in the presence of PML/RARa, therefore its function in response to ROS accumulation results is insufficient (Figure 5). 

## 5. Conclusions

Resistance to ATRA/ATO treatment is rare but grave. Treatment with high doses of ASC, directly targeting thePML/RARa protein and causing at the same time severe redox stress, is highly effective on APL cells and harmless on HSCs. Here we describe a novel direct regulatory effect of the fusion oncoprotein on the NRF2 transcription factor and cell metabolism in APL and propose high dose ascorbate treatment for APL patients who are resistant to therapy.

## Figures and Tables

**Figure 1 cancers-12-00095-f001:**
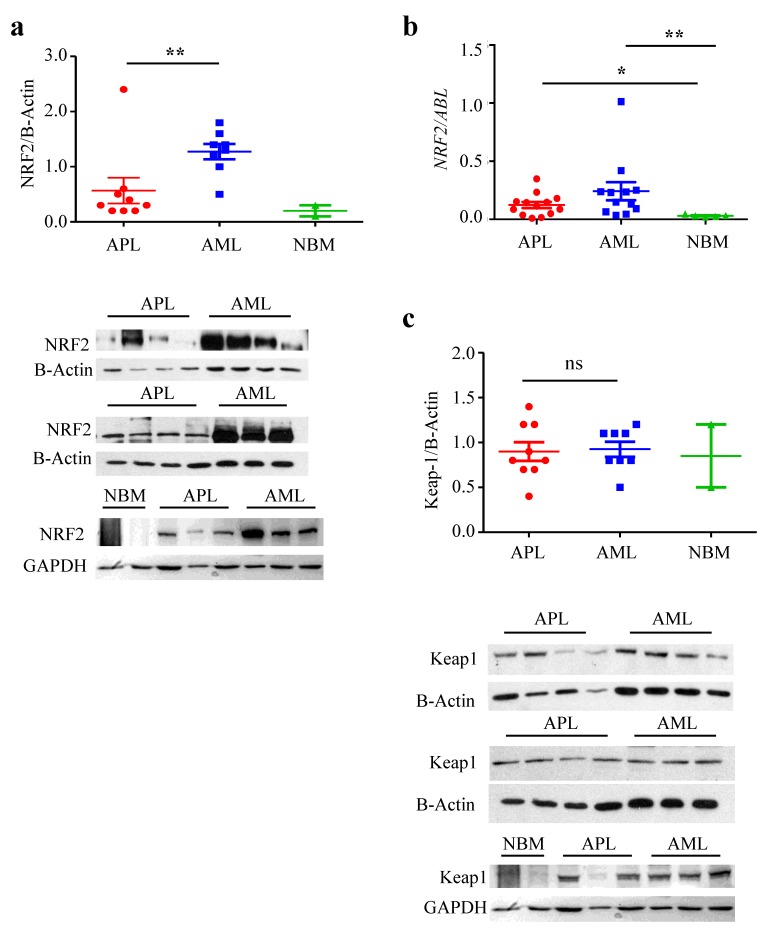
NF-E2 p45-related factor 2 (NRF2) protein level is lower in acute promyelocytic leukemia (APL) than in other acute myeloide leukemia (AML). (**a**) Western blot analysis of NRF2 in two samples from normal bone marrow (NBM), nine samples from APL patients and eight samples from AML patients. (**b**) Q-RT-PCR on NRF2 mRNA in 13 APL, 12 AML and five normal bone marrow (NBM) samples. (**c**) Western blot analysis of Keap1 in two samples from NBM, nine samples from APL patients and eight samples from AML patients. ns: non significative *: *p* ≤ 0.05; **: *p* ≤ 0.005 by Mann Withney test, not normal distribution.

**Figure 2 cancers-12-00095-f002:**
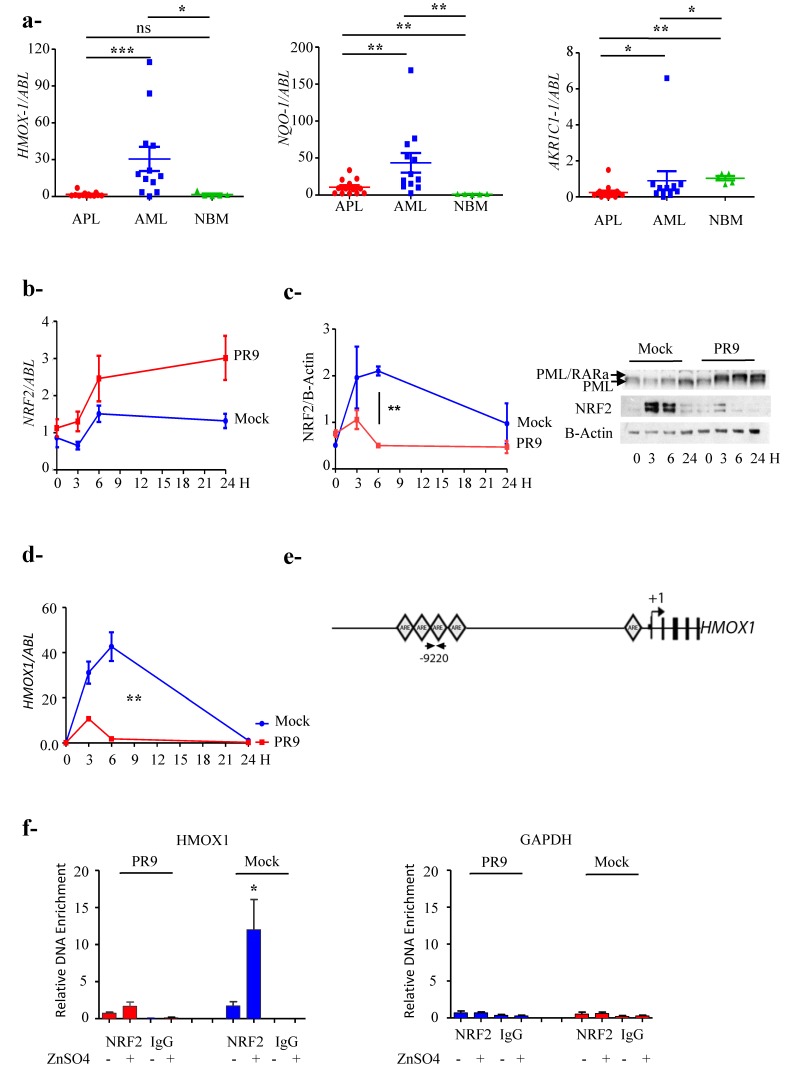
Promyelocytic leukemia/retinoic acid receptor α (PML/RARa) inhibits NF-E2 p45-related factor 2 (NRF2) transcriptional activity by preventing its binding to antioxidant response elements (ARE) motifs. (**a**) mRNA expression of NRF2 target genes: *HMOX1*, *NQO-1* and *AKR1C-1* in 13 APL, 12 AML and 5 normal bone marrow (NBM) samples. * *p* ≤ 0.05; **: *p* ≤ 0.01; ***: *p* ≤ 0.001 by the Mann–Whitney test, not normal distribution. (**b**) NRF2 mRNA expression induction in a 24 h time course after ZnSO4 addition is higher in PR9 than in Mock control cells. The experiments were performed in triplicate. (**c**) NRF2 protein level increases and persists higher in a 24 h time course after treatment with ZnSO_4_ in Mock control cells; conversely, in PR9 cells after a short lived augment, it abates due to the presence of PML/RARa. The experiments were done by triplicate. **: *p* ≤ 0.005 by unpaired *t*-test. (**d**) PML/RARa expression after treatment with ZnSO_4_ in PR9 cells inhibits the transcriptional activity of NRF2 as measured by *HMOX1* expression. The experiments were done by triplicate. **: *p* ≤ 0.005 by unpaired *t*-test. (**e**) The squares indicate the position of putative ARE motifs relative to the *HMOX1* transcription start site (TSS). The arrows indicate the position of primers used to analyze the putative ARE binding site in the HMOX1 gene. The experiment were done by triplicated. *: *p* ≤0.05 by unpaired *t*-test. (**f**) PML/RARa decreases binding of NRF2 to the *HMOX1* promoter region. The binding of NRF2 markers to DNA was measured by a quantitative ChIP assay in PR9 and Mock cells after treatment with ZnSO_4_ (100 µM). Data are shown as fold-enrichment of ChIP DNA versus input DNA. GAPDH was used as negative control. Data are representative of four independent experiments. *p* = 0.02 by Mann–Whitney test. Not normal distribution.

**Figure 3 cancers-12-00095-f003:**
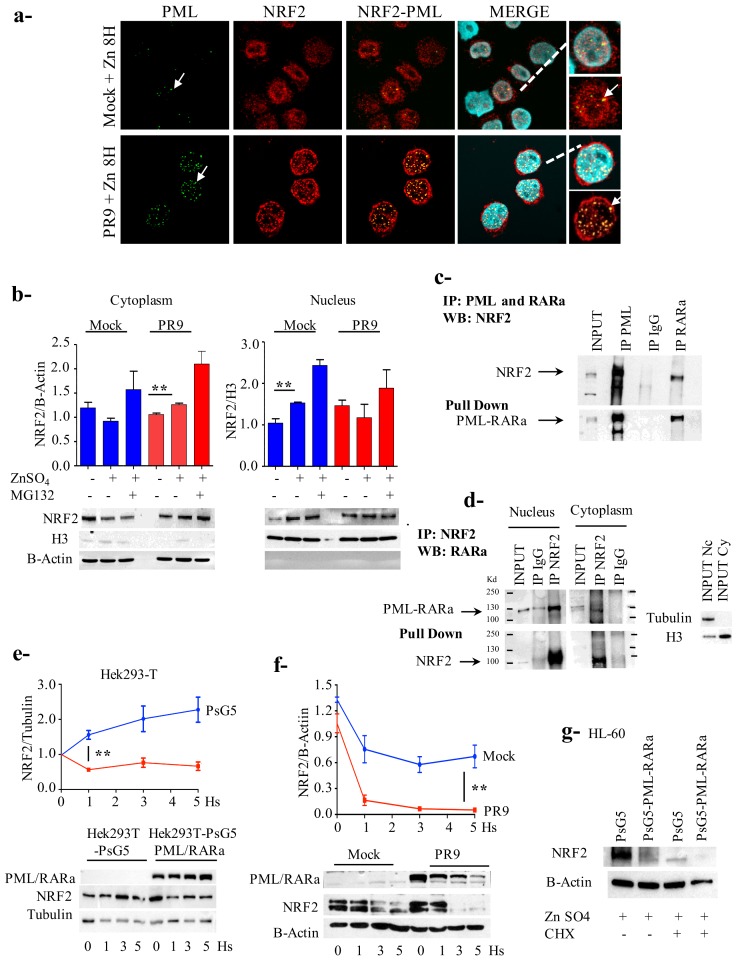
Promyelocytic leukemia/retinoic acid receptor α (PML/RARa) binds to NF-E2 p45-related factor 2 (NRF2), promotes its cytoplasmic degradation and shortens its lifespan. (**a**) At confocal microscopy NRF2 translocates to the cytoplasm eight hours following PML/RARa expression induction. The arrow indicates promyelocytic leukemia-nuclear bodies (PML-NBs). (**b**) Cytoplasmic translocation is confirmed by nuclear/cytoplasmic fractionation. (**c**) PR9 cells where treated with ZnSO_4_ 100 μM for four hours, after that cross-immunoprecipitation experiments where done showing that PML/RARa interacts with NRF2. (**d**) Co-immunoprecipitation of NRF2 in the cytoplasm and nucleus using PR9 cells previously treated with ZnSO_4_ for four hours demonstrated an interaction between PML/RARa and NRF2 in both fractions. (**e**) Hek293T cells were transfected with pS5G or pS5G-PML/RARa, after 24 h where treated with ZnSO_4_ 100 μM to evaluate the effect of PML/RARa on NRF2 induction. PML/RARa inhibit the expression of NRF2 (**f**) PR9 cells and the control Mock cells were treated with ZnSO_4_ 100 μM for two hours and then with cycloheximide (CHX) 100 μg/mL for five hours. Time 0 refers to the moment in which cycloheximide was added, +2 h after zinc induction. PML/RARa reduce significantly the half-life of NRF2. (**g**) HL60 cells were transfected with pS5G or pS5G-PML/RARa for 24 h, then treated with ZnSO4 100 μM for two hours and then with CHX 100 μg/mL for four hours. PML/RARa reduce the half life of NRF2. This experiment was done twice. All the other experiments were done by triplicate. *: *p* ≤ 0.05; **: *p* ≤ 0.01 by unpaired *t*-test.

**Figure 4 cancers-12-00095-f004:**
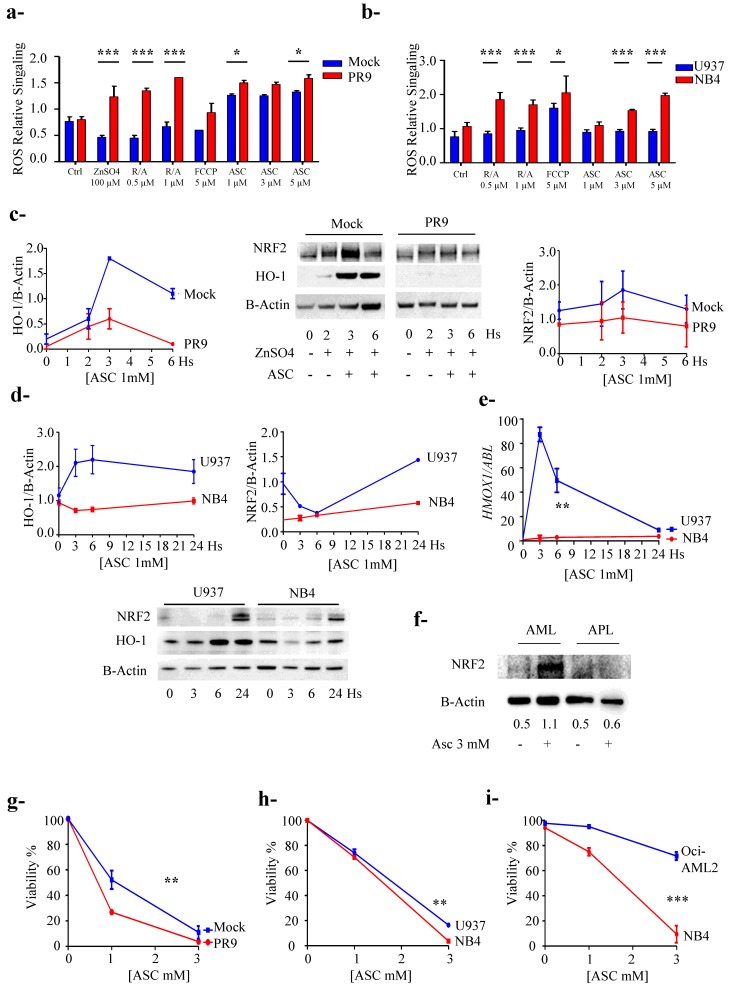
Promyelocytic leukemia / retinoic acid receptor α (PML/RARa) expression sensitizes cells to ascorbate treatment. (**a**) The levels of reactive oxygen species (ROS) content in PR9 and Mock cell lines systems were measured after induction of PML/RARa with ZnSO4. ROS levels were also evaluated after treatment with rotenone/antimycin A, Carbonyl cyanide 4-(trifluoromethoxy)phenylhydrazone (FCCP) and ASC as oxidants. In all systems the presence of PML/RARa induced a significantly higher level of ROS production. (**b**) The level of ROS production was evaluated in NB4, bearing PML/RARa, and U937 cells as control using ZnSO4, Rotenone/Antimicyn A, FCCP and ascorbate (ASC) as oxidant. NB4 cells produced higher level of ROS compared to U937 cells (**c**) PR9 and Mock cell lines systems were treated with ASC 1 mM for six hours and NRF2 and HO-1 protein expression was evaluated by western blot. In PR9 cells the PML/RARa presence inhibited the expression of NRF2 and its target HO-1 protein. (**d**) NB4 and U937 cells were treated with 1 mM ASC and evaluated for NRF2 and HO-1 production by western blot. In NB4 cells PML/RARa presence inhibited NRF2 protein and downregulated its target HO-1. (**e**) NB4 and U937 cells were treated with ASC 1 mM for 24 h and *HMOX1* mRNA expression was measured demonstrating NRF2 transcriptional deficiency. (**f**) Primary blasts from one AML and one APL patient were treated with ASC 3 mM for 24 h: NRF2 was clearly abated in the APL blasts. (**g**) PR9 and Mock control cells were treated with ZnSO4 100 μM for two hours then with Asc at 1 and 3 mM for 72 h and vitality was assessed by MTS assay. (**h**) NB4 and U937 cells were treated with increasing concentrations of ASC and viability was assessed using the ATP lite test. (**i**) NB4 and Oci-AML2 cells were treated with increasing concentrations of ASC and viability was assessed by cytometric analysis using Anexin – PI. *: *p* < 0.05, **: *p* < 0.01, ***: *p* < 0.001 by unpaired *t*-test. All the experiments were performed in triplicate.

**Figure 5 cancers-12-00095-f005:**
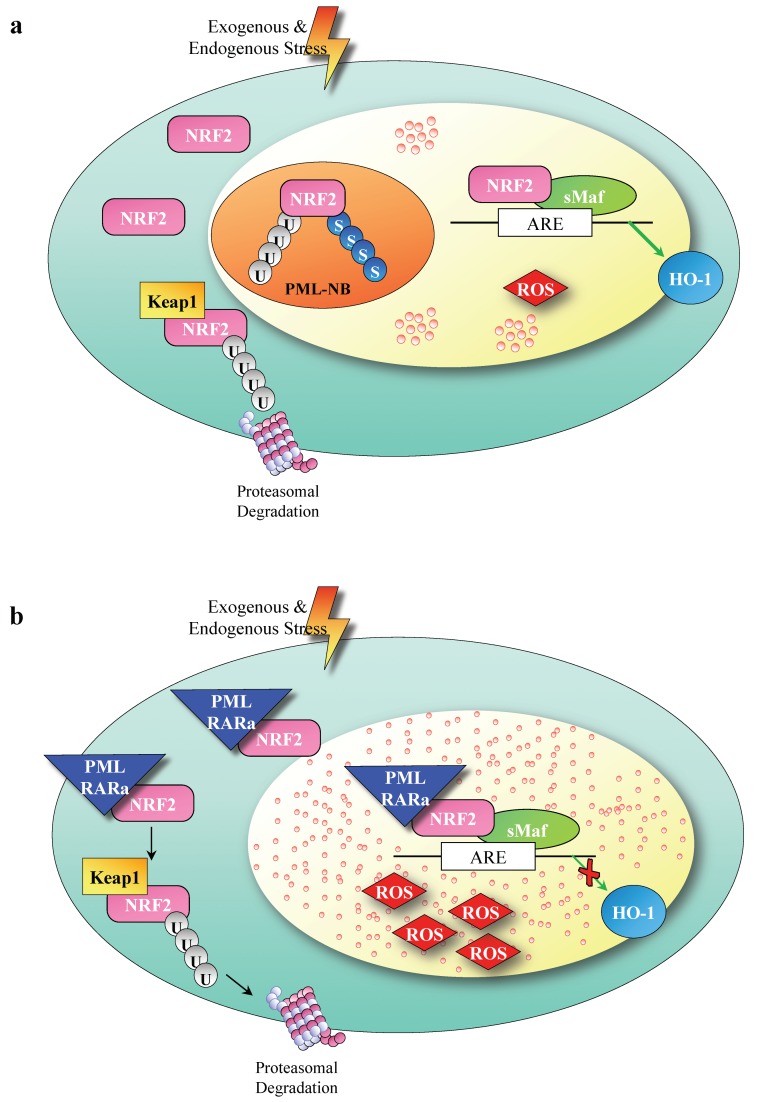
A model of NF-E2 p45-related factor 2 (NRF2) deregulation by Promyelocytic leukemia/retinoic acid receptor α (PML/RARa). (**a**) NRF2 activity is maintained at a low level via Keap1 binding, constitutive ubiquitination and proteosomal degradation. Electrophiles and oxidants inhibit its degradation, enabling NRF2 protein to accumulate in the nucleus initiating a genetic program to allow cellular adaptation to stress. (**b**) PML/RARa binds to NRF2 protein, segregates it to the cytoplasm, accelerates its degradation, and impedes its upregulation and induction of (antioxidant response elements) ARE-driven genes in response to electrophiles and oxidants; therefore, NRF2 function in response to ROS accumulation is insufficient.

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
