# Peer review of "PML/RARa Interferes with NRF2 Transcriptional Activity Increasing the Sensitivity to Ascorbate of Acute Promyelocytic Leukemia Cells"

_cancers, 2019, doi:10.3390/cancers12010095_

Round 1

Reviewer 1 Report

The authors have addressed all my comments satisfactorily.

Reviewer 2 Report

In this study, the authors reported that PML/RARa in APL cells inhibits NRF2 function, impedes the translocation to nucleus and enhances degradation in the cytoplasm. As a result, increased amount of ROS and more sensitivity against ascorbic acid may be induced. The aim of this paper is very important in understanding of molecular mechanism of PML/RARAa in APL and the development of therapy against APL. The data and descriptions in Fig 3d of second version of paper are resolved by additional experiments and descriptions.

Reviewer 3 Report

The authors did not perform all the experiments requested by the reviewer, but the manuscript is substantially improved though this revision.  

This manuscript is a resubmission of an earlier submission. The following is a list of the peer review reports and author responses from that submission.

Round 1

Reviewer 1 Report

This is a very interesting paper with potentially a very high impact. Overall, the data are sound and support the conclusions. There are some minor points that should be addressed.

In Figure 3b, it is not clear what difference there is between the + symbols beneath the x-axis - are these increasing concentrations of ZnSO4?

In Figure 3, two panels are both labelled as Figure3c.

In Figure 4a, although there is a clear difference in ROS response between the Mock and PR9 cells exposed to ZnSO4, in both cases ROS signalling appears lower than in untreated cells. Can the authors explain this? Is there a clear induction of ROS at an earlier time-point? If so, this data should be included.

The text on line 352 should be modified, since "..PML/RARa bearing NB4 and U937 cell lines..." suggests that both cell lines are positive for PML-RARa.

In Figure 4c, it appears that the legend is incorrect. The text states that HMOX1 transcript induction is inhibited in NB4 cells - which is consistent with the rest of the paper - but the graph shows induction in NB4 cells and suppression of induction in U937 cells.

Author Response

Specific comments:

REV 1

This is a very interesting paper with potentially a very high impact. Overall, the data are sound and support the conclusions. There are some minor points that should be addressed.

In Figure 3b, it is not clear what difference there is between the + symbols beneath the x-axis - are these increasing concentrations of ZnSO4?

We thank you for your comments. The figure was also incomplete, lacking the mention of MG132 treatment. It was corrected.

In Figure 3, two panels are both labelled as Figure3c.

Since we added some more data the figure was adjourned and corrected.

In Figure 4a, although there is a clear difference in ROS response between the Mock and PR9 cells exposed to ZnSO4, in both cases ROS signalling appears lower than in untreated cells. Can the authors explain this? Is there a clear induction of ROS at an earlier time-point? If so, this data should be included.

We thank you for your comments. There is an increase in ROS production in the first minutes since the product is chemically an oxidant. But since we sought to ascertain PML/RARa effect on NRF2 we had to wait two hours for the induced transcript to reach a critical quantity as to have a remarkable effect, at that time point the cellular homeostatic mechanism has overwhelmed the ROS unbalance caused by ZnSo4 with a rebound effect. Since more than one reviewer commented about the figure we changed the experiment adding more oxidants and revising the conditions, the new, more complete results, are summarized in a revised fig 4.

We added to the text the following paragraph: ‘To evaluate NRF2 inhibition by PML/RARa on the oxidative stress defense mechanism of the cell we measured the levels of ROS content in NB4 and U937 cell lines and PR9 and Mock cell lines systems after induction with ZnSO4. As mentioned before Zinc sulphate induces oxidative stress, and enhances NRF2 protein content, but in PR9 cells induces PML/RARa expression as well. To further explore the oxidative unbalance we added respectively 0.5 and 1 µM of rotenone/antimycin A and 5 µM Carbonyl cyanide p-(tri-fluromethoxy)phenyl-hydrazone (FCCP) and growing concentration of ASC (1,3 and 5 mM), to induce oxidative stress. In Mock cells Zinc sulphate addition induces NRF2 expression and ROS cellular content remains stable, but PML/RARa expression in PR9 cells with consequent NRF2 loss of function causes significant augment of ROS (Fig. 4a). Challenged with the other oxidants both cell lines system showed that NRF2 promptly responds to the challenge when is not present PML-RARa, conversely its presence gravely impairs NRF2 homeostatic function (Figure 4a, 4b).

To confirm that NRF2 inhibition participate to the mechanism of the enhanced sensitivity to Ascorbate of APL cells we treated the two different cellular systems with ASC 1 mM and measured NRF2 and HO-1 protein expression and found that in the presence of PML/RARa downgrade NRF2 protein quantity and downregulate HO-1 protein (Fig. 4c, 4d). To make sure that the effect we sow was due to transcriptional deficiency of NRF2 we measured HO-1 mRNA levels in the NB4/U937 system by RT-PCR after treatment with ASC 1mM for 24 hours. We observed a clear inhibition of HMOX1 transcript induction in NB4 cells as compared with control cells (2.3 ± 3.2 vs 87.3 ± 8.2 at 3 hours and 2.9 ± 2.9 vs 59.4 ± 13.8 at 6 hours) p=0.005 (Fig. 4e). To confirm our finding in a natural occurring system we then challenged with ASC 3 mM for 24 hours two sets of fresh primary blast, one from an APL patient and one from a patient with a different AML subtype, and confirmed NRF2 downgrade in the presence of PML/RARa (Figure 4f).

We had proof that both in cell lines and fresh blasts from patients the presence of PML/RARa inhibits the expression of NRF2 target genes, that should sensitize cells to treatment with ASC 1 mM. In accordance with our data showing inhibition of NRF2 activity by PML/RARa, we observed that induction of PML/RARa NB4 and sensitized induced PR9 cells to ASC by, respectively, ATP Lite and MTT viability tests, registering an enhancement of its efficacy in the presence of PML/RARa protein (p= 0.04 and 0.03) (Figure 4g, 4h).’ Lines 408-436.

The text on line 352 should be modified, since "PML/RARabearing NB4 and U937 cell lines..." suggests that both cell lines are positive for PML-RARa.

The construction was corrected (see the paragraph above)

In Figure 4c, it appears that the legend is incorrect. The text states that HMOX1 transcript induction is inhibited in NB4 cells - which is consistent with the rest of the paper - but the graph shows induction in NB4 cells and suppression of induction in U937 cells.

We thank you for your comments. The figure was corrected.

Reviewer 2 Report

In this study, the authors reported that PML/RARa in APL cells inhibits NRF2 function, impedes the translocation to nucleus and enhances degradation in the cytoplasm. As a result, increased amount of ROS and more sensitivity against ascorbic acid may be induced. The aim of this paper is very important in understanding of molecular mechanism of PML/RARAa in APL and the development of therapy against APL. However there are some unclear or unsuitable data and descriptions in this paper. Criticisms regarding this paper are discussed below.

Comments

The selection of leukemic cells in analysis of NRF2 function.

The authors used “NB4 vs U937” and “Mock vs PR9” as leukemic cell lines with/without PML/RARa gene. AML cell lines without t(15;17) are appropriate to the aim because U937 cell is known as monocytoid cell line derived from histiocytic lymphoma. Please examine other cell lines such as HT93 cell line with t(15;17) (Exp Hematol. 26:135, 1998) and other AML cell lines without t(15;17) transfering of PML/RARa transgene in order to secure the reproducibility.

ZnSO4 as only one exogenous stress.

The authors should examine other stress such as oxidents in order to observe NRFs alterations in Fig 2, 3, and 4 since the stimulus by zinc is one of peculiar stresses.

Protein analysis using western blot

There are some bands PML/RARa in MOCK cells in Fig 2-c. What are these bands ?

The bands of PML/RARa in PR9 cells at 3H, 6H and 24H seem to be higher than at 0H. Why increased moleculer number of PML/RARa protein at 3H, 6H and 24H ?  

Fig 3 is confusing.

It seems NRF2 proteins locate in both cytoplasm and nucleus in Fig 3-a. Please explain this image.

Please indicate nuclear body in Mock cells using arrow if there is.

“Mock 8H” may be corrected “Mock + Zn 8H” in Fig 3-a ?

There are two Fig 3-c (IP blot and NRF2 in Hek293T-PsG5). Please repair.

What is smear-like bands of NRF2 in IP with anti-PML antibody ?

Please indicate vertical axis, NRF2, in Fig 3-c and d.

ROS analysis and effect of ascorbic acid.

The examination of ROS measurement and alteration of HMOX1 expression / cell viability after ascorbic acid treatment should be done using both “NB4 vs U937” and “Mock vs PR9” as least in Fig 4-a, b, c and d.

Please examine altered expression of NRF2 after ascorbic acid treatment in APL cells and AML cells in cases similar to Fig 1 and then in “NB4 vs U937” and “Mock vs PR9”.

The authors summarize PML/RARa protein locates in cytoplasm associate with NRF2 protein in Fig 5-B. However it seems PML/RARa may be almost in nucleus in Fig 3-a. It should be examined by IP of NRF2 and PML/RARa in cytoplasm / nucleus.

Author Response

REV 2

In this study, the authors reported that PML/RARa in APL cells inhibits NRF2 function, impedes the translocation to nucleus and enhances degradation in the cytoplasm. As a result, increased amount of ROS and more sensitivity against ascorbic acid may be induced. The aim of this paper is very important in understanding of molecular mechanism of PML/RARAa in APL and the development of therapy against APL. However there are some unclear or unsuitable data and descriptions in this paper. Criticisms regarding this paper are discussed below.

Comments

The selection of leukemic cells in analysis of NRF2 function.

The authors used “NB4 vs U937” and “Mock vs PR9” as leukemic cell lines with/without PML/RARa gene. AML cell lines without t(15;17) are appropriate to the aim because U937 cell is known as monocytoid cell line derived from histiocytic lymphoma. Please examine other cell lines such as HT93 cell line with t(15;17) (Exp Hematol. 26:135, 1998) and other AML cell lines without t(15;17) transfering of PML/RARa transgene in order to secure the reproducibility.

We thank you for your comments. To make sure that the effect we were seeing was affecting a general cellular mechanism and not just something peculiar of the myeloid environment, we tested also a non myeloid cell line, we transfected Human Embryo Kidney Hek293 cells with PsG5-PML/RARa or control PsG5 constructs and results confirmed the result obtained using the PR9 – Mock system (Figure 3 e, lines: 361-366). According with your suggestion, since we could not have the cell line you mentioned in a suitable time for responding to the journal, we transfected an ulterior myeloid cell line: HL60. We added the results to figure 3g and the following line was added to the text: ‘A Similar result was obtained treating the myeloid HL-60 cell line transfected with PsG5-PML/RARa or control PsG5 constructs with ZnSO4 for two hours and then with CHX (100μg/ml) for four hours. In the control cells NRF2 expression is much higher respect to PML/RARa transfected cells, the difference persists after CHX treatment (Figure 3g)’ lines 372-379.

ZnSO4 as only one exogenous stress.

The authors should examine other stress such as oxidents in order to observe NRFs alterations in Fig 2, 3, and 4 since the stimulus by zinc is one of peculiar stresses

We thank you for your comments. Since more than one reviewer commented about the figure 4 we changed the experiment adding more oxidants and revising the conditions, the new, more complete results are summarized in a revised fig 4.

We added to the text the following paragraph: ‘‘To evaluate NRF2 inhibition by PML/RARa on the oxidative stress defense mechanism of the cell we measured the levels of ROS content in NB4 and U937 cell lines and PR9 and Mock cell lines systems after induction with ZnSO4. As mentioned before Zinc sulphate induces oxidative stress, and enhances NRF2 protein content, but in PR9 cells induces PML/RARa expression as well. To further explore the oxidative unbalance we added respectively 0.5 and 1 mM of rotenone/antimycin A and 5  mM Carbonyl cyanide p-(tri-fluromethoxy)phenyl-hydrazone (FCCP) and growing concentration of ASC (1,3 and 5 mM), to induce oxidative stress. In Mock cells Zinc sulphate addition induces NRF2 expression and ROS cellular content remains stable, but PML/RARa expression in PR9 cells with consequent NRF2 loss of function causes significant augment of ROS (Fig. 4a). Challenged with the other oxidants both cell lines system showed that NRF2 promptly responds to the challenge when is not present PML-RARa, conversely its presence gravely impairs NRF2 homeostatic function (Figure 4a, 4b).’ Lines 408-419.

Protein analysis using western blot

There are some bands PML/RARa in MOCK cells in Fig 2-c. What are these bands ?

The bands of PML/RARa in PR9 cells at 3H, 6H and 24H seem to be higher than at 0H. Why increased moleculer number of PML/RARa protein at 3H, 6H and 24H ?

The figure is confusing.

We thank you for your comments. We used anti PML antibody who recognize both PML and PML-RARa. We corrected the figure and indicated those bands that are PML bands with lower molecular weight, and are present in all the lanes, and the higher PML-RARa specific bands that are present just at 3,6 and 24 hours after induction in PR9 cells.

Fig 3 is confusing.

It seems NRF2 proteins locate in both cytoplasm and nucleus in Fig 3-a. Please explain this image.

Biological phenomena are never absolute clear cut when observed at a distance as in a microscopy essay, the point of the images is that there is a perceptible higher quantity of NRF2 protein in the subtle cytoplasmic rim in PR9 treated cells than in untreated PR9 and treated Mock cells and that some agglomerates show a yellow coloration due to optical wave length interference caused by strict proximity of large quantity of the two proteins

Please indicate nuclear body in Mock cells using arrow if there is.

We indicated PML- NBs with an arrow.

“Mock 8H” may be corrected “Mock + Zn 8H” in Fig 3-a ?

The figure has been corrected

There are two Fig 3-c (IP blot and NRF2 in Hek293T-PsG5). Please repair.

Since we added some more data the figure was adjourned and corrected.

What is smear-like bands of NRF2 in IP with anti-PML antibody ?

Some antibody stain more than others, especially when a relatively large quantity of protein is loaded in the gel well, the smear like image is an artifact due to the protein quantity and the antibody physical behavior, since is very hard to preconize the quantity of protein pulled down by immunoprecipitation and one has to make sure that all the bands, even the weaker, are visible sometime there are overloaded lanes like that.

Please indicate vertical axis, NRF2, in Fig 3-c and d.

The figure has been corrected, previous figures 3c and d now are labelled as figures 3e and 3f.

ROS analysis and effect of ascorbic acid.

The examination of ROS measurement and alteration of HMOX1 expression / cell viability after ascorbic acid treatment should be done using both “NB4 vs U937” and “Mock vs PR9” as least in Fig 4-a, b, c and d.

The experiments were performed and results summarize in figure 4a- 4h. The following paragraph was added to the text: ’To confirm that NRF2 inhibition participate to the mechanism of the enhanced sensitivity to Ascorbate of APL cells we treated the two different cellular systems with ASC 1 mM and measured NRF2 and HO-1 protein expression and found that in the presence of PML/RARa downgrade NRF2 protein quantity and downregulate HO-1 protein (Fig. 4c, 4d). To make sure that the effect we sow was due to transcriptional deficiency of NRF2 we measured HO-1 mRNA levels in the NB4/U937 system by RT-PCR after treatment with ASC 1mM for 24 hours. We observed a clear inhibition of HMOX1 transcript induction in NB4 cells as compared with control cells (2.3 ± 3.2 vs 87.3 ± 8.2 at 3 hours and 2.9 ± 2.9 vs 59.4 ± 13.8 at 6 hours) p=0.005 (Fig. 4e). To confirm our finding in a natural occurring system we then challenged with ASC 3 mM for 24 hours two sets of fresh primary blast, one from an APL patient and one from a patient with a different AML subtype, and confirmed NRF2 downgrade in the presence of PML/RARa (Figure 4f).

We had proof that both in cell lines and fresh blasts from patients the presence of PML/RARa inhibits the expression of NRF2 target genes, that should sensitize cells to treatment with ASC 1 mM. In accordance with our data showing inhibition of NRF2 activity by PML/RARa, we observed that induction of PML/RARa NB4 and sensitized induced PR9 cells to ASC by, respectively, ATP Lite and MTT viability tests, registering an enhancement of its efficacy in the presence of PML/RARa protein (p= 0.04 and 0.03) (Figure 4g, 4h).’ Lines 420-436.

Please examine altered expression of NRF2 after ascorbic acid treatment in APL cells and AML cells in cases similar to Fig 1 and then in “NB4 vs U937” and “Mock vs PR9”.

The experiment were performed (see the paragraph above).

The authors summarize PML/RARa protein locates in cytoplasm associate with NRF2 protein in Fig 5-B. However it seems PML/RARa may be almost in nucleus in Fig 3-a. It should be examined by IP of NRF2 and PML/RARa in cytoplasm / nucleus.

We performed immunoprecipitation on fractioned nuclear and cytoplasmic extracts. The following paragraph was added to the manuscript: ‘Since PML/RARa protein localize mostly in the nucleus according to microscopy (Figure 3a) we investigated if there was a direct physical interaction between PML/RARa and NRF2 in the cytoplasm. We performed co-immunoprecipitation essay on fractioned nuclear and cytoplasmic extracts using anti-NRF2 antibody and challenged the blot with anti-RARa antibody, demonstrating a much larger amount of complexed PML-RARa/NRF2 proteins in the cytoplasmic fraction (figure 3d)’ lines .355-360’

Reviewer 3 Report

This study examined relationship between PML/RARa and NRF2, but lacks sufficient explanation through the manuscript. Unfortunately, this manuscript is too premature for publication. 

The introduction does not provide sufficient background or explain why the authors decide to examine the relationship. In addition, the methods and explanation of results and legends are not well described. There are many error through the manuscript such as labeling and grammar. Most of the data are not convincing. 

Author Response

REV 3

This study examined relationship between PML/RARa and NRF2, but lacks sufficient explanation through the manuscript. Unfortunately, this manuscript is too premature for publication. 

The introduction does not provide sufficient background or explain why the authors decide to examine the relationship. In addition, the methods and explanation of results and legends are not well described. There are many error through the manuscript such as labeling and grammar. Most of the data are not convincing. 

We thank you for your remarks. It is difficult to respond since you do not contextualize your criticisms in the text. We try to summarize what were our intention in working on PML/RARa effect on NRF2 protein. The study addresses a precise physical interaction between oncoprotein PML-RARa and NRF2 protein with demonstrated biological effect and significance and repercussion on therapeutic aspects. On the ground of previously published experiments concerning an innovative therapy for acute myeloid leukemia using high doses of ascorbate we searched for a physiological mechanism who could explain the peculiar sensitivity of APL cells to the therapy. Since NRF2 is known to orchestrate cellular response to stress we essayed NRF2 protein in leukemic blasts and found it downregulated in APL cells. We demonstrate unquestionably a general cellular mechanisms acting in APL blasts, using fresh blast cells and six different cell lines (one non myeloid to demonstrate independence from the myeloid environment). We corrected labelling errors and typos.

Round 2

Reviewer 2 Report

In this study, the authors reported that PML/RARa in APL cells inhibits NRF2 function, impedes the translocation to nucleus and enhances degradation in the cytoplasm. As a result, increased amount of ROS and more sensitivity against ascorbic acid may be induced. The aim of this paper is very important in understanding of molecular mechanism of PML/RARAa in APL and the development of therapy against APL. The unsuitable data and descriptions in first version of paper are almost resolved by some additional experiments and descriptions.

Minor comment

In figure 3d, there is a smear-like broad band (of lower MW than PML-RARalpha ptotein) of WB with anti RAR alpha antibody using IP with anti NRF2 antibody in the nucleus fraction. What is this band ? Please explain these proteins in this paper.

Reviewer 3 Report

The authors do not understand what the reviewer pointed out in the first review, therefore detailed points are indicated below.

Ascorbate is key word in this study, but there is no explanation about what is ascorbate in introduction. There is only mention that high doses of ascorbic acid preferentially kill leukemic blast from APL patients in previous study.

In Figure 1, expression of Nrf2 and Keap1 in APL and AML were examined, but there is no explanation why the authors first decided to examine Nrf2 expression in these cells.

There is no explanation what is AML in this manuscript. AML may be acute myelogenous leukemia. Relationship between PML/RARa and AML should be explained, but there is no explanation in this study. In Figure 1, expression of PML/RARa should be shown.

Nrf2 mRNA was examined in NBM as well as APL and AML in Figure 1b, but Nrf2 protein was examined only in APL and AML in Figure 1a. Nrf2 protein in NBM should be examined in Figure 1a. In addition, number of examined blasts was different between Figure 1a and 1b.

Association of protein level and mRNA level of Nrf2 among each cells of APL, AML and NBM should be shown. In addition, association of Nrf2 target genes such as HMOX1, NQO1 and AKR1 (shown in Figure 2a) and Nrf2 protein levels in each cell also should be shown.

In Figure 1c, Keap1 expression also should be examined in NBM as well as APL and AML.

Keap-1 should be Keap1 in figure 1c, figure 1c legend and text (line250).

Line 274, “100 uM” should be “100 µM”.

In Figure 3a, Zn-induced PML/RARa is shown to localize mostly in nucleus not in cytoplasm. Therefore, interaction between PML/RARa and Nrf2 in cytoplasm is not convincing.

In Figure 3b, H3 expression in cytoplasm fraction of Mock cells seems to be higher than that of PR9 cell, indicating that cytoplasm/nucleus fractionation is not fair between Mock and PR9.

In Figure 3b, blot of Nrf2 in cytoplasm faction of PR9 is not clear. This is key point in this study, therefore more convincing data should be shown.

In Figure 3b, graph and band intensity for Nrf2 in nucleus fraction of PR9 is not consistent.

In Figure 3c, pull-down Nrf2 should be shown for upper panel. In the same way, pull-down PML/RARa should be shown for lower panel. In Figure 3d, pull-down Nrf2 should be shown.

In legend for Figure 3. It is mentioned “PML/RARa directly binds to NRF2”. The data presented in this study cannot exclude indirect interaction between PML/RARa and NRF2. Direct interaction should be examined by using of recombinant proteins of PML/RARa and NRF2.

Interacting domains of PML/RARa and NRF2, respectively should be examined. INot only PML/RARa fusion protein but also non-fusion proteins of PML and PAR should be examined for interaction with Nrf2.

It is weird that PML/RARa expression is detected even without ZnSO induction (0 Hs) in Figure 3e.

In Figure 3e, Nrf2 protein level in 0 Hs of Hek293T-PsG5 (lane 1) appears lower than that in 0 Hs of Hek293T-PsG5-PML/RARa (lane 5). It is not fair.

In Figure 3g, Tubulin band intensity appears weak, indicating loading amount between lanes are not equal.

Line 351, “A Similar result“ should be “A similar result”.

Line 392, “the effect we sow” may be “the effect we saw”

In Figure 4c, band size of Nrf2 in PR9 is different between 0 Hs and ZnSO4 treated condition.

In Figure 4c, quantitative graph for Nrf2 protein should be shown.

In Figure 4d, error bar is not shown for graph of HO-1 expression.

In Figure 4g, difference of effect of ASC on viability between PR9 and Mock is small. Also, in Figure 4h, difference of effect of ASC on viability between U937 and NB4 is small.

Moch in Figure 4g and U937 in Figure 4h should be same, but there is huge difference of effect of ASC on viability. How come this discrepancy can be explained.